# Stem Cell Therapy for Modulating Neuroinflammation in Neuropathic Pain

**DOI:** 10.3390/ijms22094853

**Published:** 2021-05-03

**Authors:** Hari Prasad Joshi, Hyun-Jung Jo, Yong-Ho Kim, Seong-Bae An, Chul-Kyu Park, Inbo Han

**Affiliations:** 1Department of Neurosurgery, School of Medicine, CHA University, CHA Bundang Medical Center, Seongnam-si 13496, Gyeonggi-do, Korea; hariprasadjoshi10@gmail.com (H.P.J.); anseongbae@gmail.com (S.-B.A.); 2Spinal Cord Research Centre, Regenerative Medicine Program, Department of Physiology and Pathophysiology, Rady Faculty of Health Sciences, University of Manitoba, Winnipeg, MB R3E 0W2, Canada; 3Gachon Pain Center, Department of Physiology, College of Medicine, Gachon University, Incheon 21999, Gyeonggi-do, Korea; hj840318@hanmail.net (H.-J.J.); euro16@gachon.ac.kr (Y.-H.K.)

**Keywords:** mesenchymal stem cell, neural stem cell, neuropathic pain, neuroinflammation, neuromodulation

## Abstract

Neuropathic pain (NP) is a complex, debilitating, chronic pain state, heterogeneous in nature and caused by a lesion or disease affecting the somatosensory system. Its pathogenesis involves a wide range of molecular pathways. NP treatment is extremely challenging, due to its complex underlying disease mechanisms. Current pharmacological and nonpharmacological approaches can provide long-lasting pain relief to a limited percentage of patients and lack safe and effective treatment options. Therefore, scientists are focusing on the introduction of novel treatment approaches, such as stem cell therapy. A growing number of reports have highlighted the potential of stem cells for treating NP. In this review, we briefly introduce NP, current pharmacological and nonpharmacological treatments, and preclinical studies of stem cells to treat NP. In addition, we summarize stem cell mechanisms—including neuromodulation in treating NP. Literature searches were conducted using PubMed to provide an overview of the neuroprotective effects of stem cells with particular emphasis on recent translational research regarding stem cell-based treatment of NP, highlighting its potential as a novel therapeutic approach.

## 1. Introduction

Pain is the body’s response to unpleasant (noxious) stimuli, such as external injury or internal disease. The physiological response to pain is essential for the body’s risk awareness and risk avoidance [1]. Chronic pain is defined as pain that persists after healing is, or exists without tissue damage, and usually lasts longer than three months [2]. The prevalence of chronic pain increases through adult life, and previous estimates of chronic pain in the adult population ranged from 11% to 19% [1]. Neuropathic pain (NP) is defined by The International Association for the Study of Pain (IASP) as pain that occurs as a direct consequence of lesion or disease affecting the somatosensory system [2]. NP makes up 20–25% of chronic pain patients, and although it may vary, due to a global dissonance of the definition of NP, a systematic epidemiological review has estimated the prevalence of NP between 3% and 17% [3]. Despite significant advances in NP treatment, safe and effective treatment options targeting NP are lacking. As a novel treatment, stem cell-based therapy is gaining significant attention. In this review, the possibilities of stem cell use in NP patients and relevant challenges in their use have been discussed. We searched the keywords of “Neuropathic pain”, “Stem cell”, and “Neuroinflammation” in PubMed (https://pubmed.ncbi.nlm.nih.gov/, accessed on 28 February 2021).

## 2. Chronic NP

NP can be classified depending on the underlying lesion or disease [2], or according to the clinical phenotype [3]. The 11th edition of the International Classification of Diseases (ICD-11) distinguishes NPs of peripheral and central origin, consisting of nine common conditions associated with persistent or recurrent pain [3,4] (Table 1). Chronic NP caused by a lesion or disease in the somatosensory nervous system can be spontaneous and cause an increased response to painful stimuli (hyperalgesia) or a painful response to painless stimuli (allodynia) [3]. The subtypes of chronic peripheral NP are the following: Trigeminal neuralgia, chronic NP after peripheral nerve injury, painful polyneuropathy, postherpetic neuralgia, and painful radiculopathy. The following forms belong to chronic central NP: Chronic central NP associated with spinal cord injury (SCI), chronic central NP associated with brain injury, chronic central poststroke pain, and chronic central NP associated with multiple sclerosis [1,2] (Figure 1). NP may result from a broad range of systemic nervous disorders affecting the peripheral or central nervous system, which can be etiologically classified as mechanical, metabolic, ischemic, inflammatory, neurotoxic, radiation-associated, or hereditary [5] (Table 2). 

In general, chronic NP is poorly recognized, poorly diagnosed, and poorly treated. Diagnosis of chronic NP requires a medical history of lesion or disease of the nervous system and clinical examination showing negative (e.g., decreased or loss of sensation) or positive sensory signs (e.g., allodynia or hyperalgesia) with a plausible neuroanatomical distribution [3]. NP treatment is a real challenge for physicians. NP management primarily targets clinical symptoms instead of causative factors. Currently, available treatment options include both pharmacological and nonpharmacological approaches (Table 3). Regarding pharmacological therapies of NP, tricyclic antidepressants (e.g., amitriptyline, nortriptyline), serotonin-norepinephrine reuptake inhibitors (e.g., duloxetine and venlafaxine), and gabapentinoids (i.e., gabapentin and pregabalin) are recommended as first-line drugs. Weak opioid analgesics, such as tramadol, are recommended as second-line drugs. Topical medications (such as lidocaine plaster and capsaicin patch) are recommended as second-line pharmacological treatments only in peripheral NP. As third-line drugs, strong opioids, such as morphine and oxycodone, are recommended both in central and peripheral NP conditions. In trigeminal neuralgia, carbamazepine and oxcarbazepine are the first drugs of choice [6]. Nonpharmacological treatment options for drug-refractory NP include the following approaches: Interventional therapies (e.g., peripheral nerve blockade, epidural steroid injection, sympathetic nerve/ganglion blockade, intrathecal morphine delivery, and peripheral and central neurostimulation), physical therapies (e.g., massage, ultrasound, transcutaneous electrical nerve stimulation), and psychological therapies, such as cognitive behavioral therapy [6].

## 3. Role of Immune Cells in NP

Immune system activation plays a major role in both peripheral and central abnormal sensory processing. While NP has long been thought to arise from neurons, recent studies have highlighted the immune response’s important role in the development of NP. Immune cells are not only the source of pain mediators, but also produce analgesic molecules [7] (Figure 1). Mast cells were activated in a model of partial sciatic nerve injury [8]. Further, Behrooz et al. [9] indicated that mast cell transplantation could enhance the functional recovery of transected sciatic nerves. The endoneurial accumulation of neutrophils at sites of peripheral nerve injury is important in the early pathogenesis of hyperalgesia. Further, neuroimmune interactions occur as a result of peripheral nerve injury and are important in the subsequent development of NP [10].

Therefore, neutrophils could play a major role in early NP development. Several studies have indicated that macrophages were implicated in the pathogenesis of allodynia or hyperalgesia [11]. Macrophages play a major role in Wallerian (axon) degeneration, a characteristic of chronic constriction-associated painful neuropathy, through the release of pro-inflammatory cytokines at the site of nerve damage [12]. Further, Wallerian degeneration is a key factor in the pathogenesis of hyperalgesia. After a nerve lesion, macrophages infiltrate the area of Wallerian degeneration. Intravenous injection of clodronate encapsulated in liposomes reduced the number of macrophages in the injured nerves, alleviated thermal hyperalgesia, and protected myelinated, as well as unmyelinated fibers against degeneration [13]. In parallel to or following macrophage recruitment, T cells also infiltrate into damaged nerves. Chronic constriction injury (CCI) rats exhibited considerable macrophage and T cell infiltration both at the site of injury, as well as at the dorsal root ganglia (DRG, a cluster of sensory neuron cell bodies). Further, pain and disability rat models had significantly upregulated T cell numbers [14]. Several studies have demonstrated that neuroinflammation resulting from the activation of microglia and astrocytes plays a critical role in the development and maintenance of NP [15,16,17,18]. Activated microglia and astrocytes release a variety of proinflammatory cytokines, as well as chemokines after injury, including IL-1β, IL-6, and TNF-α, which are the most extensively studied proinflammatory mediators. These are not only involved in astrocyte-microglia crosstalk, but are also implicated in inflammation-associated pain, bone cancer pain, and NP. Spinal microglia, which respond to extracellular stimuli, facilitate signal transduction through intracellular cascades, such as mitogen-activated protein kinase, p38, and extracellular signal-regulated protein kinase signaling [16]. Astrocytes are more closely associated with chronic pain behavior and the synapse following nerve injury, exhibiting a more persistent reaction than microglia [17]. Thus, microglia might be responsible for the initiation of NP, while astrocytes are implicated in its persistence [19].

Alexander et al. [20] indicated that pain following nerve damage can be mitigated by cytotoxic natural killer cells that selectively clear out partially damaged nerves. Nerve injury triggers an organized cascade of events to mount an inflammatory response [21]. Immediately after injury, glial cells surrounding the nerve are activated, releasing cytokines and chemokines within minutes, driving neutrophil recruitment. Neutrophils are usually the first peripheral immune cells to invade the injury site. Monocyte-derived macrophages infiltrate damaged nerves within hours to days. Thereafter, T cells usually arrive at the site of injury and distal parts of the nerve, the DRG, and finally, infiltrate the dorsal horn of the spinal cord. Moalem et al. [22] described the kinetics of T cell infiltration in the sciatic nerve in response to CCI in rats. T cells were not observed in sham and contralateral nerves, and only a few T cells were detected at three days after injury. Significant T cell infiltration at proximal and distal sites of injury was observed at seven days, reaching a peak at 21 days. Infiltrated T cells were still detected 40 days after injury (the last time point checked) [22]. This pattern is consistent with several studies using different nerve injury models of rats and mice, wherein few T cells were observed at the site of injury at three days postsurgery, and their number significantly increased between 7 and 28 days postsurgery [23].

## 4. Inflammation and Pain

Inflammation is a major biological process with numerous roles beyond the response to infection. The inflammatory process is central in various diseases, including cancer and diabetes. It is increasingly recognized that the immune system interacts with the sensory nervous system, contributing to persistent pain states [24,25].

Pain generation and transmission occur via synapses and neurotransmitter release between neurons. Nociceptors are a subset of primary afferent neurons whose cell bodies are located in the DRG and trigeminal ganglia. These neurons respond to tissue damage and consist of both unmyelinated C-fibers, as well as myelinated Aδ-fibers that innervate skin, muscle, joints, and visceral organs [26]. Nociceptor excitation in the cerebral cortex occurs through signal transduction, transmission, and modulation. Tissue damage causes an inflammatory response in nociceptors, which may lead to their excitation, generating an action potential, which is transmitted through the axon to excite neuronal bodies in the abdominal dorsal root ganglion. Neurons of the abdominal dorsal root ganglion propagate the potential through axons extending toward the spinal cord, with the signal reaching nerve endings of the spinal cord located within the dorsal horn. As a result, neurotransmitters are released at synapses to stimulate secondary neurons and transmit pain signals to the brain [27].

Peripheral sensitization is caused by the activation of various ion channels, including transient receptor potential ion channels (e.g., TRPA1, TRPV1, and TRPM8) [28], sodium channels (e.g., Nav1.7, Nav1.8, and Nav1.9) [29], and mechanosensitive PIEZO ion channels [30]. Further, pain-mediating neurotransmitters are divided into inflammatory (e.g., prostaglandins, prostacyclin, leukotrienes, adenosine triphosphate, adenosine, substance P, proton (H^+^), nerve growth factors, 5-hydroxytryptamine, histamine, glutamate, neurokinin, nor-epinephrine, and nitric oxide) and noninflammatory (e.g., calcitonin gene-related peptide, γ-aminobutyric acid, opioid peptides, glycine, and cannabinoids) [27]. All of these mediators bind and activate their cognate receptors located on postsynaptic neurons. As a result, pain sensation is amplified through various secondary messengers. For instance, (cAMP)/protein kinase A (PKA) and protein kinase C (PKC)/diacylglycerol (DAG) signaling have been reported as important for maintaining peripheral hyperalgesia, whereas cyclic guanosine monophosphate (cGMP) had a negative effect on cAMP signaling during nociceptor sensitivity [31,32]. Peripheral sensitization was enhanced in response to TNF-α, as indicated by an increase in TRPV1 activity. Similarly, Nav1.8 activity was upregulated in response to IL-1β. Both of these ion channel responses resulted from p38 MAPK activation in DRG neurons [33,34].

## 5. The Alternative/Nonpharmacological Treatment of Neuropathic Pain: Spinal Cord Stimulation (SCS)

Given the fact that common analgesic and opioid therapies are associated with non-negligible risk of adverse events in the long term, lesional surgery at the dorsal root entry zone, and more recently, a number of neuromodulation procedures, have been suggested as alternative options [35]. Among them, spinal cord stimulation (SCS) or dorsal column stimulation constitutes an advanced neuromodulation procedure enabling to potentially decrease neuropathic pain in many syndromes, such as in failed back surgery syndrome (FBSS), complex regional pain syndrome (CRPS) type I and II, postherpetic neuralgia, and pure radicular pain [35,36]. Despite its proven efficacy, the favorable cost-effectiveness, when compared to the long-term use of poorly effective drugs and the expanding array of indications and technical improvements, SCS is still worldwide largely neglected by general practitioners, neurologists, neurosurgeons, and pain therapists, often bringing to a large delay in considering as a therapeutic option for patients affected by chronic neuropathic pain [35,37]. Over the last years, a growing number of chronic pain syndromes of neuropathic origin have been treated with SCS, from brachial plexus and peripheral nerve injuries to central pain of spinal cord origin, with varying grades of evidence [38] (Table 3). SCS is usually regarded as a safe procedure, due to its reversible and minimally invasive characteristics. Severe adverse events, such as spinal epidural bleeding and permanent neurologic deficit, are rare, whereas hardware complication and infection have been reported with an incidence of 24–50% and 7.5%, respectively [35,36,37] (Table 3). Although still underused, conventional SCS may be considered as an effective, safe, well-tolerated, and reversible treatment option for severe drug-refractory neuropathic pain. Accurate indications and cautious patient selection represent the principal mainstays for the success of this treatment. In the near future, there will surely be confirmations as to the efficacy of the new patterns of stimulation both at high frequency and through burst stimulation, and possibly, future new patterns to improve the efficacy of this treatment in improving chronic neuropathic pain.

## 6. Stem Cell Therapy

Over the last decade, stem cell transplantation has exhibited remarkable potential for the repair of nervous system damage in NP syndromes rather than simply providing temporary palliation. Stem cell therapy has thus emerged as an alternative therapeutic approach for NP [50]. In particular, human mesenchymal stem cells (hMSCs) have been characterized as neuroprotective in spared nerve injury (SNI) [51]. Mechanistically, stem cells represent a totipotent cellular source, replacing injured or lost neural cells. Further, they provide trophic factors to the injured nerve. Herein, we review the available literature on diverse stem cell types implicated in neural repair and NP treatment, focusing on their mechanism of action, while also addressing their advantages and limitations. Finally, we suggest future directions for these treatment strategies.

Among stem cell types explored for the mitigation of neurological diseases, neural stem cells (NSCs) are considered the superior option, due to their higher capacity for differentiation into neurons, oligodendrocytes, and astrocytes [52,53]. They help to bridgethe injury gap and facilitate lesion repair. NSCs are primarily derived from the olfactory bulb, hippocampal dentate gyrus, embryos, the neonatal and adult spinal cord of rodents, as well as the human fetal CNS, the SVZ surrounding ventricles, and the corpus callosum [54]. Reports suggested that sciatic nerve CCI-induced peripheral neuropathic pain was significantly attenuated by the intravenous delivery of NSCs [55]. Thus far, NSCs have been considered to treatperipheral NP. Interestingly, recent reports have suggested that they are equally effective to treat SCI-mediated NP [56]. In an earlier study, NSC administration decreased both the mRNA and protein levels of proinflammatory IL-1 at the lesion site, thereby significantly attenuating hyperalgesia [54]. In addition, the pain-relieving effect of NSCs has been suggested to occur through a reduction of spinal cord Fos expression. As NSCs possess extensive self-renewal capacity, they retain the ability to generate mature functional brain cells over time. In a previous study, even though stem cell transplant efficiency was low, cells were present at the lesion site from day 1 to day 7 after injection. NSCs were also isolated from the SVZ using the neurosphere technique [57]. In another study, the authors carried out intrathecal NSC administration in a rat model, which resulted in the significant reduction of mechanical and thermal hyperalgesia [58]. MSCs are an extensively studied heterogeneous subset of stromal stem cells with great potential in pain management research. They comprised of the progenitor cell population of mesodermal origin found within the bone marrow of adults, giving rise to skeletal muscle, blood, adipose tissue, vascular, urogenital, and connective tissue throughout the body [59,60]. Based on their origin, MSCs can be sub-divided into bone marrow-derived MSC (BM-MSCs), adipose tissue-derived MSC (AD-MSCs), as well as umbilical cord blood-derived stem cells (UCB-MSCs) and amniotic mesenchymal stem cells, which have gained increasing attention in recent years. Research has shown that MSCs can be sourced from the dental pulp, placenta, fetal liver, and lungs [61,62]. MSCs exhibit high expansion potential, genetic, as well as phenotypic stability, and can be easily collected and shipped from the laboratory to the bedside, while also being compatible with different delivery methods and formulations [63]. In addition, MSCs have two advantageous characteristics: They can migrate to sites of tissue injury and have strong immunosuppressive properties that can be exploited for successful autologous or heterologous transplantation [64]. The molecular mechanisms through which MSCs exert their beneficial effects with regard to pain are yet to be clarified. However, a previous study reported that MSCs migrated to injured tissue and mediated functional recovery following brain, spinal cord, and peripheral nerve lesions, suggesting that these cells could modulate pain generation following sciatic nerve constriction [65]. Further, transplanted stem cells have been demonstrated to ameliorate disease symptoms through the integration of new graft-derived cells, which provide trophic support to endogenous cells and facilitate immunomodulation (Figure 2) [66,67]. In addition to that, MSCs show their immunomodulatory effects via secreted cytokines and growth factors via direct cell interactions, as well as strong paracrine influences. MSCs secrete biological factors via extracellular vesicles (EVs), which are divided into microvesicles (>200-nmdiameter) and exosomes (50–200 nm diameter) [68]. Extracellular vesicles are composed of thousands of proteins, messenger RNA, and/or microRNA [69], many of which are reported to enhance neuronal growth and health in model systems [70]. Furthermore, MSCs can induce upregulation of regulatory T cells, which are thought to play a key role in amyotrophic lateral sclerosis (ALS) [71] (Figure 2).

In an earlier study, a detailed analysis of hMSC treatment effects in NP was carried out by using an experimental mononeuropathy pain mouse model. Reduction in NP following stem cell transplantation was monitored based on pain-like behavior analysis (thermal hyperalgesia and mechanical allodynia). It was also previously reported that single ligature sciatic nerve constriction-mediated mechanical and cold allodynia were significantly suppressed by ipsilateral intraganglionic injection of bone marrow stromal cells (BMSCs) in rats [65]. The possible mechanism behind this effect could be that BMSCs partially prevent the injury-induced galanin, neuropeptide Y, and neuropeptide Y receptor expression in DRG [72]. Maione et al. were the first to use hBMSCs in an NP model. They employed a spared nerve injury (SNI) mouse model and injected hBMSCs four days after surgery via the caudal vein and cerebral ventricle [73]. As a result, mechanical allodynia and thermal hyperalgesia were greatly alleviated. Further, glial and microglial activation were found to be downregulated along with suppression of proinflammatory cytokines and the upregulation of their anti-inflammatory counterparts [73].

Among the diverse MSC categories, AD-MSCs obtained from the mature subcutaneous tissue are advantageous over the other MSC types. While these cells exhibit low immunogenicity and possess considerable immunomodulatory properties, their use in experimental NP studies has received little interest. The antinociceptive effect of human ASCs (hASCs) isolated from the adipose tissue was first reported by Sacerdote et al. in 2013 [74]. Intravenous injection of 1 × 10^6^ hASCs completely mitigated thermal hyperalgesia. In another study, hASCs were found to be more effective than NSCs [75]. The antinocicetive effects of hASCs were due to their anti-inflammatory properties, as IL-10 upregulation was observed, while IL-1 was downregulated after treatment. Choi et al. [76] Incorporated hASCs in ZNO cell nanoparticles, and they were successfully transplanted.

Human UB-MSCs (hUB-MSCs) and Wharton’s jelly-derived MSCs (WJ-MSCs) have gained attention within the regenerative medicine field. They both have advantages, including: (1) Noninvasive collection procedures for allogeneic and autologous transplants; (2) very low infection rate, (3) immunosuppressive capability and low immunogenicity, (4) lower risk of teratomas [77]. Previous research revealed that the combined use of hUB-MSCs and human amniotic epithelial stem cells (hAESCs) led to a significant attenuation of mechanical allodynia [77].

Similarly, a recent study suggested that NP attenuation could be achieved using bone marrow-derived mononuclear cells (BM-MNCs). Klass et al. [78] suggested that the intravenous injection of rat BM-MNCs significantly reduced NP10 days after rat CCI. BM-MNCs isolated from were able to reduce STZ-induced diabetic neuropathy in rats [79]. Briefly, the cells were injected into the hind limb skeletal muscles, and significant amelioration of mechanical hyperalgesia and cold allodynia was observed on the BM-MNC-injected side two weeks later. Furthermore, the decreased sciatic nerve blood flow and slowed sciatic nerve conduction velocities (MNCV/SNCV) of diabetic rats were greatly improved on the BM-MNC-injected side, along with suppressed NT-3 expression and a lower number of hind limb’smicrovessels (Table 4).

### 6.1. Stem Cell Mechanism of Action in the Pain Recovery Process

#### 6.1.1. Effect of Stem Cells in Peripheral NP

##### Anti-Inflammatory Regulation

Following nerve injury, peripheral sensitization leads to the infiltration of immune cells, such as neutrophils, macrophages, and mast cells, at the injury site causing overexcitation and continuous discharge of nerve fibers [80]. Following inflammation, a large number of cytokines, chemokines, and lipid mediators are released, sensitizing and stimulating nociceptors, which in turn results in local homeostatic changes [80]. Reports from preclinical animal models suggested that anti-inflammatory cytokines exerted analgesic effects [81]. The immunomodulatory and angiogenic properties of stem cells have also been reported [82,83]. In a CCI sciatic nerve injury model, both IL-1β and IL-6 expression were greatly attenuated following transplantation of adipose-derived stem cells, whereas, anti-inflammatory factor IL-10 was significantly upregulated [84]. These outcomes could be due to the interaction between stem cells and macrophages, leading to polarization of the latter into anti-inflammatory phenotypes. This hypothesis is further supported by another study wherein dental pulp stem cell (DPSC) transplantation in diabetic rats led to an increase in M2 macrophages. Further, upregulatedM2 macrophage markers were also observed in RAW264.7 cells in vitro [85].

Similarly, damaged axons have been reported to secrete factors that activate the extracellular signal-related MAPK signal pathway in Schwann cells, which is crucial for inflammatory regulation (Figure 3) [86]. There are four major MAPK-related pathways, namely, the ERK1/2, JNK, p38, and ERK5 cascades. Among these, ERK1/2 and p38 play a central role in pain modulation. Following intrathecal injection of BMSCs in a rat CCI model, DRG pERK1/2 expression was greatly reduced in DRG [87].

#### 6.1.2. Effect of Stem Cells in Spinal NP

##### Attenuation of Central Sensitization

Nerve injury leads to a series of neuronal cascades, which result in central sensitization, the crucial step in NP development. As shown in Figure 4, among the diverse neurotransmitters and pain-associated molecules released at the synaptic connections in the dorsal horn of the spinal cord, glutamate plays a major role by binding to its inotropic glutamate receptor, thus triggering central sensitization.

Following nerve injury, significant amounts of excitatory neurotransmitter glutamate are continuously released in the synaptic junctions of the spinal dorsal horn, activating the glutamate receptor (Glu-R), *N*-methyl-d-aspartate (NMDA) receptor (NMDAR), and α-amino-3-hydroxy-5-methyl-4-isoxazolepropionic acid receptor (AMPA receptor, AMPAR). Receptors then transmit pain signals to sensory brain regions [88,89,90]. Studies in various animal models have confirmed that blocking NMDAR can relieve NP [91]. Specific NMDAR antagonists have been used intermittently for NP [1]. Guo et al. intravenously injected BMSCs into tendon ligation (TL) and SNL rat models. They observed that BMSCs could inhibit the expression of NMDARs and protect rats from glutamate excitotoxicity, alleviating mechanical hyperalgesia after SCI and demonstrating the beneficial analgesic properties of stem cells for chronic pain [92].

Studies have also demonstrated that TGF-β1 attenuates glutamate-induced excitotoxic neuronal damage in rat neocortical neurons in a concentration-dependent manner [93]. TGF-β1 is a well-known neuroprotective and neurotrophic factor with an active role in synaptic transmission. Following brain injury, TGF-β1 regulates the excitatory synaptic transmission of spinal cord neurons through the TGF-β1 receptor. BMSC transplantation in a mouse neuralgia model up regulated TGF-β1 levels in cerebrospinal fluid. In addition, higher TGF-β1 levels were observed in the BMSC culture medium compared to the normal culture medium. To determine whether TGF-β1 was involved in the antinociceptive effect of BMSCs in NP, mice were treated with a specific neutralizing antibody against TGF-β1 at three days after BMSC injection. TGF-β1 neutralization reversed the antihyperalgesic effect of BMSCs [94]. Moreover, intrathecally injected BMSC in CCI-induced mice showed an increase in TGF-β1 secretion along with antiallodynic and antihyperalgesic effects, while IL-10, other anti-inflammatory cytokines, released from BMSC were very low and did not contribute to alleviating allodynia and hyperalgesia [94]. These findings indicated that stem cells could suppress the increase of neuronal excitability after nerve injury by releasing TGF-β1, resisting central sensitization, and thus, exerting an analgesic effect.

##### Inhibition of Glial Cell Activation

Glial cells account for approximately 70% of central nervous system cells and play an important role in maintaining homeostasis (Figure 4) [91]. Major glial cells include astrocytes, oligodendrocytes, and microglia [95]. A previous study demonstrated that astrocytes accumulate immediately after injury and last for 12 weeks, whereas microglia appear within 24 h [96]. Glial cells release inflammatory cytokines resulting in glutamate receptor upregulation and pain hypersensitivity [91,96]. Stem cells are known to suppress glial cell activation. In our previous study, GFAP and IBA-1 were significantly expressed in DRG and spinal cord samples from CCI rats [18,97]. Interestingly, following ADSC transplantation, GFAP expression was greatly decreased [97]. Similarly, in a rat model of disc herniation, microglial activity in the spinal cord dorsal horn was greatly suppressed after BMSC transplantation [98].

Another study suggested that glial activation upregulates MAPK signaling, which in turn promotes long-term potentiation and central pain sensitization [99]. Stem cells have been proven as capable of inhibiting microglial activation and MAPK signaling in activated glial cells. BMSC transplantation was previously reported to decrease the activation of p-p38MAPK and p-ERK1/2 in microglia following SCI in rats while improving their motor function [100]. It is suggested that stem cells not only inhibit the activation of microglia and astrocytes directly, but also suppress microglial activation by inhibiting astrocyte CCL7 secretion [101]. However, the mechanisms of stem cell and glial cell interactions with regard to pain require further investigation.

### 6.2. Viral Vector-Mediated Gene Transfer into Stem Cells

Viral vector-mediated gene therapy is promising in clinical trials in nervous system diseases. Glutamic acid decarboxylase (GAD) enzyme, a key enzyme in the biosynthesis of GABA, plays an important role in the analgesic mechanism. Therapeutic gene transfer may locally produce neurotransmitters/neuropeptides while avoiding unwanted side effects that would cause activation of the same receptors in other locations/pathways by a systemically administered drug [102]. Using viral vectors to express target gene products could represent an alternative to standard pharmacological approaches. Gene therapy is a promising choice to treat many central nervous system (CNS) disorders in clinical trials, including chronic pain [102]. Subcutaneous inoculation of HSV vector expressing GAD67 in rats with PDN reduced mechanical hyperalgesia, thermal hyperalgesia, and cold allodynia, and prevented the increase in the voltage-gated sodium channel isoform 1.7 (NaV1.7) protein [103]. Studies have shown that viral vectors, including HSV, AV, AAV, and HFV, express GAD67 or GAD65 for chronic pain treatment. These studies provide promising support for the clinical trial using viral vector-mediated GAD for therapeutic options of chronic pain.

Considering their regenerative ability, gene modification of stem cells has several advantages over conventional gene therapy. Ex vivo gene transfection of stem cells may avoid the administration of vectors and vehicles into the recipient organism. A number of requirements for gene transfer to stem cells determine the choice of the appropriate vector or gene transfer vehicle. Retroviral vectors most reported attempts to transduce HSCs for gene therapy protocol have used retroviral vectors, such as the murine leukemia virus (MLV) [104,105]. There is increasing evidence that lentivirus-based systems might be ideal vectors for the transduction of human HSCs [106]. Adeno-associated viral vectors Recombinant adenovirus-associated viruses (rAAV), which can take up to 4.8 kb of exogenous DNA are also being explored as potential vectors for the introduction of genes into HSC [107]. Adenovirus vectors have been successfully used for transient gene expression in many systems, although standard adenovirus does not usually enable stable integration [108]. It is utilized to achieve transient genetic engineering of stem cells, especially for the acceleration of regenerative responses, such as NSCs for neurogenesis.

#### Factors Limiting Gene Transfer into Stem Cells

One important issue which potentially limits gene transfer of retroviral vectors into HSCs is the quiescent nature and reduced receptor expression of primate HSCs. Successful gene therapy applications require optimized strategies to increase gene transfer efficiency and expansion in balance with the maintenance of the immature state of HSCs. Today, important experimental variables include the multiplicity of infection, length of time or viral incubation, medium used for viral incubation, the viral construct (including promoter and gene), and the source of the stem cells. The determination of factors controlling HSC proliferation, differentiation, and expression of the transgene after each cycle of transduction are important issues that are currently under investigation. One final issue concerns the selection of successfully transduced stem cells. Ex vivo selection involves cell surface antigen expression and autofluorescent (GFP) marking, while an in vivo selection method is based on separating transduced cells by conferring drug resistance (dihydrofolate reductase) or by selective amplifier genes (chimeric receptor) [109,110]. The in vivo selection by drug treatment after transplant also offers the possibility to selectively increase the number of primitive and mature transduced stem cells [110].

## 7. Potential Shortcomings

Although stem cell-based therapies have been shown to protect against neurodegeneration and promote neuroregeneration, there are several issues that need to be addressed. The optimal dosing for stem cell transplantation remains unknown and requires further elucidation prior to clinical trials. It was previously demonstrated that hMSCs transplanted into the rat model generated different grafts depending on cell numbers: Low numbers of transplanted hMSCs generated Nestin-expressing grafts, whereas higher numbers of transplanted hMSCs generated grafts expressing astroglial markers [111]. The number of transplanted cells also raises questions regarding cell survival. One of the aforementioned studies indicated that only 1.7% of total injected hMSCs survived [112]. In another unrelated study, no cells were detected in the animal model four weeks after stem cell transplantation. The reasons for this are unknown. Possibilities include cell death or loss of fluorescence [113]. Further, there still are unknown variables in the use of stem cells to treat neuropathic disease. Nevertheless, as with any cellular therapy approach, challenges that will need to be addressed before achieving the full therapeutic potential of stem cells remain. The source of stem cells, considerations on autologous versus allogeneic transplants, precommitment to neuronal lineages, characterization of neurotrophic factor release, and dosing requirements, among other aspects, will need to be further explored by scientists to facilitate the use of stem cells for NP treatment.

There are some challenges remaining for clinical translation in stem cell therapy for NP. First, the invasive surgical process often used in treating NP related to spinal cord injury might put the patients at too much risk [114], even though preclinical studies with invasive treatments have shown an improvement in NP [70]. Second, because autologous stem cells used in the preliminary clinical trials are obtained from patients themselves, the risk of rejection and host toxicity is negligible [115]. To make stem cells to be a therapeutic drug, the use of allogeneic expansion of stem cells in the future is necessary [87].

## 8. Conclusions

NP is a chronic heterogeneous condition of the sensory nervous system, for which no curative treatment is available. Current pharmacological therapeutics are mainly palliative, providing temporary pain relief without achieving complete recovery. Stem cells present exciting therapeutic prospects for NP. Although the exact mechanism underlying stem cell-mediated pain relief remains unclear, the current review summarizes evidence on the potential of stem cells in arresting degenerative processes and promoting the survival/recovery of nerves.

A large number of clinical trials have now been performed or are ongoing. Available preclinical and clinical data highlight the positive effects of stem cells for NP relief. A phase I uncontrolled clinical trial for chronic traumatic spinal cord injury with 14 patients in 2014, autologous BMSCs that were transplanted directly into the patient’s spinal cord injury site showed some improvement in subjects’ pain symptoms by varying degrees. The intensity of NP was also significantly and gradually improved after the first BMSC transplantation, and autologous BMSCs were safely tolerated [116,117]. Moreover, Autologous adipose MSCs clinical trials for neurotrigeminal neuralgia with 10 patients showed no systemic nor local tissue side effects, and the use of antineurotic drugs were decreased in 5 out of 9 subjects [118]. Major advantages of stem cells include the fast onset and long duration of their favorable effects. Route of administration is an important variable to be considered. While local stem cell delivery is largely used, the risks of side effects, such as tissue injury, cannot be overlooked. Systemic delivery represents an attractive option, due to its superior biodistribution, but it also presents challenges, such as passive cell entrapment within tissues. As major considerations, toxicity, cytogenic aberration, and the possible malignant transformation of stem cells should be thoroughly assessed. Even though stem cells are at the early stages of clinical use, the future of this approach is very bright owing to technological advances and an increasing body of experimental, as well as clinical evidence.

## Figures and Tables

**Figure 1 ijms-22-04853-f001:**
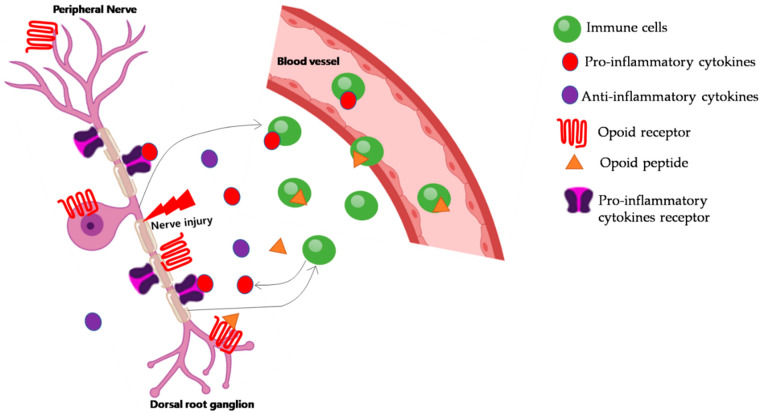
Modulation of neuropathic pain by immune cells. Following peripheral nerve injury, immune cells gather into the damaged nerve, releasing proinflammatory cytokines (e.g., TNF-α, IL-1β), which indirectly contribute to pain by interacting with their receptors. Immune cells can also produce opioid peptides, which counteract pain. Peripheral opioid receptors are expressed in dorsal root ganglia and are transported to the nerve damage site. Once there, opioid peptides activate their receptors and ameliorate neuropathic pain.

**Figure 2 ijms-22-04853-f002:**
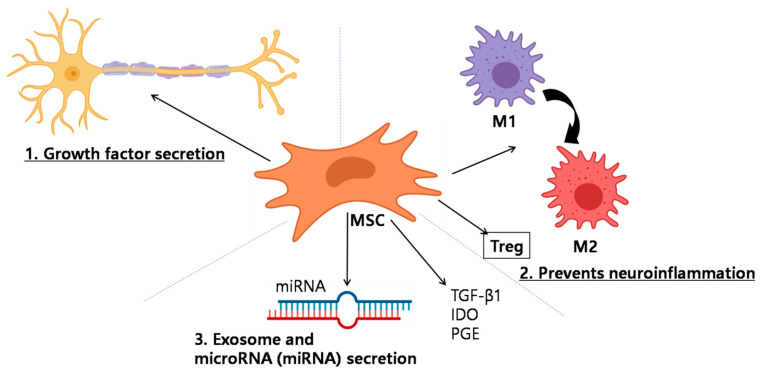
Schematic showing the mechanism of neuropathic pain recovery promoted by mesenchymal stem cells (MSCs). Multiple different mechanisms are involved: (**1**) Growth factor secretion; MSCs secrete neurotrophic growth factors, including glial cell-derived neurotrophic factor (GDNF), vascular endothelial growth factor (VEGF), and brain-derived neurotrophic factor (BDNF). Neurotrophic growth factors have been found to improve neuronal survival in neuropathic pain. (**2**) Attenuation of neuroinflammation; MSCs strongly modulate the immune system and aid wound healing. Interestingly, MSCs may be either anti-inflammatory or proinflammatory, depending on the milieu within which they exist. When entering an inflammatory milieu, MSCs become anti-inflammatory, wherein they secrete transforming growth factor β1 (TGF-β1), indole amine 2,3-dioxygenase (IDO), and prostaglandin E_2_ (PEG) and can convert macrophage/microglia from the proinflammatory M1 to the anti-inflammatory M2 phenotype. Furthermore, MSCs can induce up-regulation of T cells, which are thought to play a key role in pain regulation (**3**) exosome and microRNA (miRNA) secretion. MSCs secrete biological factors is via extracellular vesicles (EVs), such as microvesicles or exosomes. EVs are packed with thousands of proteins, messenger RNA, and/or microRNA, which have been reported to enhance neuronal growth. IDO = indole amine 2,3-deoxygenase; PGE = prostaglandin E_2_; VEGF = vascular endothelial growth factor; GDNF = glial cell-derived neurotrophic factor; TGF-β1 = transforming growth factor-β1, Treg = regulatory T cell.

**Figure 3 ijms-22-04853-f003:**
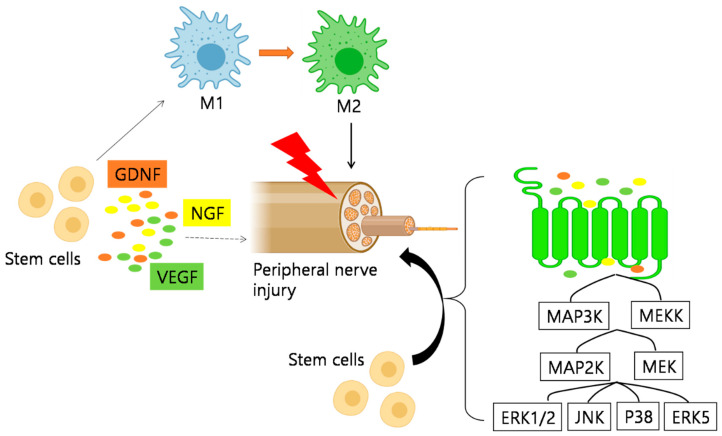
Schematic representation of the mechanism of action of stem cells in peripheral neuropathic pain. 1. Anti-Inflammatory Regulation, as stem cells promote the polarization of macrophages to anti-inflammatory phenotypes. M2 macrophages increased after MSCs treatment, while the expression of genes related to M1 macrophages decreased. 2. Neuro-protection and promotion of Axonal Myelin Regeneration. Stem cells also play an anti-inflammatory role through the mitogen-activated protein kinase (MAPK) pathway. After nerve injury, signals from damaged axons lead to the activation of the extracellular signal-related MAPK signal pathway in Schwann cells. MSCs showed to inhibit the expression of pERK1/2 in dorsal root ganglion (DRG) induced by CCI. Additionally, VEGF, GDNF, and NGF are important regulators of nerve regeneration, which can support and promote the growth of regenerated nerve fibers. GDNF = glial-derived neurotrophic factor; VEGF = vascular endothelial factor; NGF = nerve growth factor.

**Figure 4 ijms-22-04853-f004:**
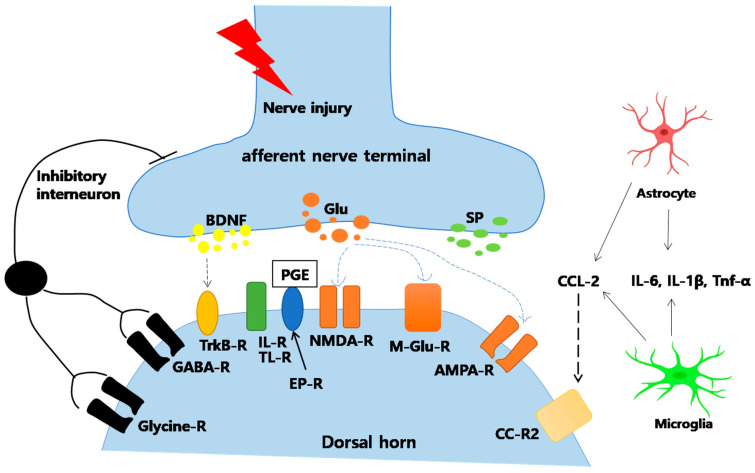
Nerve injury-associated mechanisms at the synapse between peripheral nerves and spinal cord dorsal horn neurons. 1. Weakening and Reversing Central Sensitization. After nerve injury, the release of excitatory amino acid (glutamate) in the spinal dorsal horn greatly increased, and the excitatory *N*-methyl-d-aspartate (NMDA) receptor (NMDAR) is continuously activated. Reports suggested that bone marrow stromal cells (BMSCs) could inhibit the expression of NMDA receptors and protect them from glutamate excitotoxicity, which alleviated the mechanical hyperalgesia. 2. Inhibition of Glial Cell Activation. Stem cells can effectively inhibit the activation of glial cells, such as microglia and astroglia. They also inhibit the MAPK signal pathway activation in activated glial cells. 3. Reduced Apoptosis and Autophagy of Spinal Cord Cells. The activation of intermediate inhibitory neurons leads to the release of neurotransmitter GABA, which inhibits postsynaptic neurons through membrane hyperpolarization. AMPA = *α*-amino-3-hydroxy-5-methyl-4-isoxazo lepropionic acid; BDNF = brain-derived neurotrophic factor; CCL = chemokine (C-C motif) ligand; CC-R2 = CC-chemokine receptor; DAMPs = danger-associated molecular patterns; EPR = prostaglandin E2 sensitive receptor; GABA = *γ*-aminobutyricacid; Glu = glutamate; IL = interleukin; m-Glu = metabotropic glutamate; NK = neurokinin; NMDA = *N*-methyl-d-aspartate; PAMPs = pathogen-associated molecular patterns; PG = prostaglandin; -R = receptor; SP = substance P; TLR = toll-like receptor; TNF = tumor necrosis factor; Trk = tyrosine kinase.

**Table 1 ijms-22-04853-t001:** Classification of chronic neuropathic pain in ICD-11. The specific individual concept of chronic pain is included in the following levels.

Top (1st) Level Diagnosis	Chronicneuropathic Pain	
2nd level diagnosis	Chronic peripheralneuropathic pain	Chronic central
		neuropathic pain
3rd level diagnosis	Trigeminal neuralgia Painful polyneuropathy	Chronic central neuropathic pain associated with spinal cord injury
	Chronic neuropathic pain after peripheral nerve injury	Chronic central
		neuropathic pain associated with brain injury
	Postherpetic neuralgia	Chronic central poststroke pain
	Painful radiculopathy	Chronic central
		neuropathic pain associated with multiple sclerosis
Multiple parents	Chronic posttraumatic pain	Chronic posttraumatic pain
	Chronic secondary headaches and oro-facial pain	

**Table 2 ijms-22-04853-t002:** Differential etiologies of polyneuropathies in systemic disease or conditions.

Etiology	Typical Syndrome (Example)
Mechanical	Carpal tunnel syndrome, Postsurgical pain,
	Painful radiculopathy, Cancer pain, Phantom limb pain
Metabolic/ischemic	Diabetic polyneuropathy, Vitamin B12 deficiency
Inflammatory	Postherpetic neuralgia, HIV neuropathy,
	Leprosy, Guillain-Barré Syndrome,
	Critical illness polyneuropathy
Neurotoxic	Chemotherapy-induced, peripheral neuropathy, Alcoholic neuropathy
Radiation	Postradiation neuropathy
Hereditary	Charcot-Marie-Tooth disease, Fabry disease

**Table 3 ijms-22-04853-t003:** Classification of classical pharmacological agents and alternative therapies available for neuropathic pain treatment with their mechanism of action and side effects.

Classical Pharmacological Treatment of Neuropathic Pain					
Drug Class		Types of Neuropathic Pain	Effects	Side Effects	References
Antidepressants	Tricyclic antidepressants (TCAs):	Diabetic neuropathy	Inhibition of serotonin and noradrenaline reuptake at synapses between nociceptors and spinothalamic neurons	Sedation	[39,40]
	amitriptyline, nortriptyline,	Postherpetic neuralgia		Constipation	
	desipramine, imipramine;	Poststroke pain		Weight gain	
	Serotonin-norepinephrine	Painful polyneuropathy		Dry mouth	
	reuptake inhibitors (SNRIs):	Lower back pain		Nausea	
	duloxetine, venlafaxine				
Anticonvulsants	Phenytoin	Lancinating pain and allodynia	Reduction of neuronal excitability and local neuronal discharges, acting through sodium channel blockade or modulation of calcium channels	Dizziness, Skin reaction (e.g., Steven-Johnson syndrome), Leukopenia	[41,42]
	Gabapentin	Painful diabetic neuropathy			
	Carbamazepine	Trigeminal neuralgia			
	Oxcarbazepine	Postherpetic neuralgia			
	Valproic acid	Painful polyneuropathy			
		Lower back pain			
Topical agents	Lidocaine	Allodynia	Blockade of voltage-gated sodium channels expressed by nerve fibers, responsible for the propagation of action potential.	Local irritation Possible hypersensitivity	
	Capsaicin	Postherpetic neuralgia			[43,44]
	Clonidine	Chemotherapy-induced peripheral neuropathy			
	EMLA (eutectic mixture of local anesthetic)	Postsurgical and post-traumatic neuropathic pain			
Opioids	Morphine	Diabetic peripheral neuropathy	Opioid receptors are coupled to calcium and potassium channels, block synaptic transmission, restricting the number of stimuli	Drowsiness	[39,45]
	Hydromorphine	Postherpetic neuropathy		Nausea	
	Tramadol	Polyneuropathy		Dependence overdoses	
	Oxycodone	Phantom limb pain			
Corticosteroids	Prednisolone	Allodynia	Inhibition of prostaglandin synthesis, reduction of inflammation, vascular permeability, and tissue edema	Gastrointestinal disease	[46,47]
	Dexamethasone	Spinal cord compression		psychiatric disorders	
		Postherpetic neuralgia		electrolyte imbalances	
				Bone demineralization	
Alternative nonpharmacological therapies	Acupuncture	Chemotherapy-induced peripheral neuropathy	Local inhibition of nociceptive fibers, stimulates blood flow to restore nerve damage		[48,49]
	Magnetic insoles	Trigeminal neuralgia		Bruising	
	Repetitive transcranial magnetic stimulation (rTMS)	Poststroke pain		Infection	
		Postherpetic pain			
	**Nonpharmacological/Alternative Treatment of Neuropathic Pain: Spinal Cord Stimulation (SCS)**					
	**SCS Methods**	**Types of Neuropathic Pain**	**SCS Main Contraindications**	**SCS Common Complications**	**References**
Spinal cord stimulation (SCS)	Tonic spinal cord stimulation and suprasegmental mechanisms New stimulation location: Dorsal root ganglion High-frequency spinal cord stimulation in neuropathic pain Burst spinal cord stimulation in neuropathic pain	Failed back surgery syndrome Complex regional pain syndrome (I and II) Radicular and nerve root pain Postherpetic neuralgia Pain due to peripheral nerve injury Intercostal neuralgia Phantom pain	Infection Coagulopathy Spinal stenosis Psychiatric disorders Substance abuse	More frequent: Hardware-related (lead migration, breakage, connection failure, malfunctioning, pain at the implantable pulse generator site) Hematoma and seroma at implantable pulse generator site Rare: Spinal epidural hematoma, CSF leakNeurological deficit	[35,36,37]

**Table 4 ijms-22-04853-t004:** Different types of stem cells involved in treating neuropathic pain model with their advantages and limitations.

Cell Type	Model of NP	Advantages	Limitations	Reference
Neural stem cells (NSCs)	CCI	extensive self-renewal capacity	low stem cell transplant efficiency	[55,56]
	SCI (rats)			
Mesenchymal stem cells (MSCs)	SNI (mice)	strong immunosuppressive properties; long lasting therapy	impossibility to predict which site the injected MSCs to be trapped	[61,62,66]
Bone marrow stromal cells (BMScs)	SLNC (rats)	down regulation of glial and microglial activation and proinflammatory cytokines	some limited analgesic effect	[65,72]
	SNI (mice)			
Bone marrow-derived mononuclear cells (BM-MNCs)	CCI (rats)	functional recovery of the peripheral nerve followed by increased nerve blood flow	may cause neuronal apoptosis	[78,79]
	Diabetic neuropathy (rats)			

## Data Availability

Not applicable.

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
