# Peer review of "Stem Cell Therapy for Modulating Neuroinflammation in Neuropathic Pain"

_ijms, 2021, doi:10.3390/ijms22094853_

Round 1

Reviewer 1 Report

Confidential comments to the editor:

This is a well-organized and well-illustrated paper, has an important clinical message, and should be of great interest to the readers. The review focused on the recent developments of stem cell therapy for the management of neuroinflammation and neuropathic pain. Paragraphing is concise and good, and the article consists of major recent advancements in the field of stem cell therapy for chronic pain management and deserve publication after some revisions. However, this review might need moderate English editing and the addressing some of the key questions that I have raised might improve the paper quality.

Author Response

Response to Reviewer 1: 

Comments#1: This is a well-organized and well-illustrated paper, has an important clinical message, and should be of great interest to the readers. The review focused on the recent developments of stem cell therapy for the management of neuroinflammation and neuropathic pain. Paragraphing is concise and good, and the article consists of major recent advancements in the field of stem cell therapy for chronic pain management and deserve publication after some revisions. However, this reviews might need moderate English editing and the addressing some of the key questions that I have raised might improve the paper quality.

Response#1: We greatly appreciate the reviewer’s suggestion. As suggested, we have tried to address the questions raised by the reviewers and English corrections have been made throughout the revised manuscript.

Reviewer 2 Report

Joshi  et al. present manuscript entitled Stem cell therapy for modulating neuroinflammation in neuropathic pain. Literature screening was conducted using PubMed to provide an overview of the neuroprotective effects of stem cells with particular emphasis on recent translational research regarding stem cell-based treatment of NP, highlighting its potentials a novel therapeutic approach. Review is well  structured and seems to be an important voice in academic discussion concerning neuropathic pain . 

Major

Chronic NP chapter

It is worth expanding the section of non-pharmacological treatment. Especially pay attention to the stimulation of the spinal cord which is increasingly used . Please include spinal cord stimulation (SCS) in Table 3.

Cite: https://doi.org/10.1155/2019/2606808

Reference [51] appears in text 9 times. Please find another study confirming stated thesis or rearrange. 

51 Franchi, S., et al., Adult stem cell as new advanced therapy for experimental neuropathic pain treatment. BioMed research interna- 537
tional, 2014. 2014.

Figure 1 Pictorial presentation is not clear. Please describe it in detail or rearrange.

Figure 2 description too laconic please describe presented mechanisms.

Figure 3 description too laconic please describe presented mechanisms.

Figure 4 description too laconic please describe presented mechanisms.

Check if all abreviations are explained.

Author Response

Response to Reviewer 2:

Joshi et al. present manuscript entitled Stem cell therapy for modulating neuroinflammation in neuropathic pain. Literature screening was conducted using PubMed to provide an overview of the neuroprotective effects of stem cells with particular emphasis on recent translational research regarding stem cell-based treatment of NP, highlighting its potentials a novel therapeutic approach. Review is well structured and seems to be an important voice in academic discussion concerning neuropathic pain. 

 Major

Chronic NP chapter

It is worth expanding the section of non-pharmacological treatment. Especially pay attention to the stimulation of the spinal cord which is increasingly used. Please include spinal cord stimulation (SCS) in Table 3.

Cite: https://doi.org/10.1155/2019/2606808

Response: We greatly appreciate the reviewer’s suggestion. As advised, we have expanded the non-pharmacological portion, and included the following section in the Introduction of the revised manuscript.

The alternative/ non-pharmacological treatment of neuropathic pain: spinal cord stimulation (SCS) Given the fact that, common analgesic and opioid therapies are associated with non-negligible risk of adverse events in the long term, lesional surgery at the dorsal root entry zone and, more recently, a number of neuromodulation procedures, have been suggested as alternative options [1]. Among them, spinal cord stimulation (SCS) or dorsal column stimulation constitutes an advanced neuromodulation procedure enabling to potentially decrease neuropathic pain in many syndromes such as in failed back surgery syndrome (FBSS), complex regional pain syndrome (CRPS) type I and II, postherpetic neuralgia and pure radicular pain [1, 2]. Despite its proved efficacy, the favorable cost-effectiveness, when compared to the long-term use of poorly effective drugs and the expanding array of indications and technical improvements, SCS is still worldwide largely neglected by general practitioners, neurologists, neurosurgeons and pain therapists, often bringing to a large delay in considering as a therapeutic option for patients affected by neuropathic chronic pain [1, 3]. Over the last years, a growing number of chronic pain syndromes of neuropathic origin have been treated with SCS, from brachial plexus and peripheral nerve injuries to central pain of spinal cord origin, with varying grade of evidences [4] (Table 3). SCS is usually regarded as a safe procedure due to its reversible and minimally invasive characteristics. Severe adverse events, such as spinal epidural bleeding and permanent neurologic deficit, are rare, whereas hardware complication and infection has been reported with an incidence of 24–50% and 7.5%, respectively [1, 2, 3] (Table 3). Although still underused, conventional SCS may be considered as an effective, safe, well-tolerated and reversible treatment option for severe drug-refractory neuropathic pain. Accurate indications and cautious patient selection represent the principal mainstays for the success of this treatment. In the near future, there will surely be confirmations as to the efficacy of the new patterns of stimulation both at high frequency and through burst stimulation and, possibly, future new patterns to improve the efficacy of this treatment in improving chronic neuropathic pain

Reference [51] appears in text 9 times. Please find another study confirming stated thesis or rearrange. 51 Franchi, S., et al., Adult stem cell as new advanced therapy for experimental neuropathic pain treatment. BioMed research interna- 537 tional, 2014. 2014

Response: We are thankful for the suggestion. As advised, we have added the new references and rearranged in the revised manuscript.

 Figure 1 Pictorial presentation is not clear. Please describe it in detail or rearrange.

Response: Thank you for pointing this out. The pictorial presentation has been rearranged in the revised manuscript, as suggested.

Figure 2 description too laconic please describe presented mechanisms.

Response: As advised, the mechanism has been described in the revised manuscript.

Figure 3 description too laconic please describe presented mechanisms.

Response: As advised, the mechanism has been described in the revised manuscript.

Figure 4 description too laconic please describe presented mechanisms.

Response: As advised, the mechanism has been described in the revised manuscript.

Check if all abbreviations are explained.

Response: Thank you for your request. The abbreviations are now all defined in the revised manuscript.

We greatly appreciate the reviewers’ support in strengthening this manuscript, and we thank you for considering our manuscript for review. With the above-outlined revisions, we believe that our manuscript is considerably improved and now acceptable for publication in the International Journal of Molecular Sciences. Please do not hesitate to contact me with additional comments or questions.

Thank you,

Sincerely,

In-Bo Han, M.D., PhD

Professor

Department of Neurosurgery, Spine Center

CHA University, CHA Bundang Medical Center

59, Yatap-ro, Bundang-gu, Seongnam-si, Gyeonggi-do, 13496, Korea

Tel: 031-780-1924, Fax: 031-780-5269

References

  1. Dones I, Levi V. Spinal cord stimulation for neuropathic pain: current trends and future applications. Brain sciences. 2018 Aug;8(8):138.
  2. Caylor J, Reddy R, Yin S, Cui C, Huang M, Huang C, Rao R, Baker DG, Simmons A, Souza D, Narouze S. Spinal cord stimulation in chronic pain: evidence and theory for mechanisms of action. Bioelectronic medicine. 2019 Dec;5(1):1-41.
  3. Joosten EA, Franken G. Spinal cord stimulation in chronic neuropathic pain: mechanisms of action, new locations, new paradigms. Pain. 2020 Sep;161(1):S104.
  4. Nagel, S.J.; Lempka, S.F.; Machado, A.G. Percutaneous spinal cord stimulation for chronic pain: Indications and patient selection. Neurosurg. Clin. N. Am. 2014, 25, 723–733.
